# Evaluation of Ecological Carrying Capacity and Identification of Its Influencing Factors Based on Remote Sensing and Geographic Information System: A Case Study of the Yellow River Basin in Shaanxi

Zhiyuan Zhu [1,2] , Zhikun Mei [1,2], Shilin Li [1,2], Guangxin Ren [1,2] and Yongzhong Feng [1,2,*]

1   College of Agronomy, Northwest A & F University, Xianyang 712100, China;
    zhuzhiyuan@nwafu.edu.cn (Z.Z.); zyzhu@nwafu.edu.cn (Z.M.); leeshylon@nwafu.edu.cn (S.L.);
    rengx@nwsuaf.edu.cn (G.R.)
2   The Research Center of Recycle Agricultural Engineering and Technology of Shaanxi Province,
    Xianyang 712100, China
*   Correspondence: fengyz@nwsuaf.edu.cn

**Abstract:** Ecological carrying capacity (ECC), which requires simple scientific evaluation methods, is an important evaluation index for assessing the sustainability of ecosystems. We integrate an innovative research method. Geographic information systems (GIS) and remote sensing (RS) were used to evaluate the ECC of the Yellow River Basin in Shaanxi (YRBS) and to identify the underlying factors that influence it. A calculation method that combines RS and GIS data to estimate ECC based on net primary productivity (NPP) was established. The Carnegie–Ames–Stanford approach model was applied to estimate NPP. The NPP of each land type was used as an indicator to determine the yield factors. The ECC of the watershed was calculated with the carrying capacities of each land-use type. The geographical detector model was used to study the influencing factors of ECC, which provides a scientific basis for the formulation of ecological management policies in YRBS. The results show that from 2000 to 2010, it first decreased by 45.46%, and then increased by 37.06% in 2020, an overall decrease of $13.49 \times 10^5$ wha in 20 years. Precipitation is the dominant factor that affects ECC, while the impact of human activities on ECC was significantly enhanced during the study period. The developed method based on RS data serves as a reference for ecological evaluation in other similar regions.

**Keywords:** ecological carrying capacity (ECC); Yellow River Basin in Shaanxi (YRBS); geographical detector model; remote sensing evaluation

## 1. Introduction

The ecological environment is the foundation of human existence and social development [1]. With the acceleration of industrialization and urbanization, the human consumption of natural resources and threats to the ecological environment have further increased [2,3]. At present, sustainable development, as an ideal development model, has become a policy goal that is universally recognized by all countries in the world. The 17 Sustainable Development Goals (SDGs), adopted at the United Nations Summit on September 25, 2015, aim to protect, restore, and promote the sustainable use of terrestrial ecosystems through sustainable forest management, combating desertification, halting and reversing land degradation, and eliminating biodiversity loss.

The study of ecological carrying capacity (ECC) has become an essential endeavor in several scientific disciplines including ecology and geology [4–9]. The ecological footprint method has been widely used in the study of ECC [10–12]. This method considers the productive capacity of land, which is closely related to economic, social, and technological

progress, as the regional ECC. Therefore, this method reflects the influence of natural, economic, and social systems on ECC to a certain extent [13–15].

The ecological footprint method is a biophysical evaluation method of ECC proposed and perfected by Rees. It classifies biologically productive land into six major categories based on the differences in ecosystem productivity such as fossil fuel, arable, forest, grass, construction lands, and water [16,17]. These six types of land represent mutually exclusive land uses. Owing to the different ecological productivities of these biologically productive land areas, the calculated areas of various types of land cannot be directly added. Therefore, Wachernagel used yield and equilibrium factors for these types of land areas with different ecological productivities and converted them into a composite biologically productive land area in terms of the world-average ecological productivity with a unified unit of "global hectare (gha)". In a given year, any type of biologically productive land that is converted into "global hectares" has the same ecological productivity. The various types of land areas can be summed up to indicate the total demand for regional ECC (ecological footprint) or total supply (ECC). Thus, the comparison of the ecological footprint and ECC of different countries and regions is achieved. It can be seen that yield and equilibrium factors are important indicators for the accurate calculation of ECC. However, when using the ecological footprint method, in most cases, the calculation of the equilibrium and output factors of different land types is mainly based on limited statistical results and experience, and domestic studies are comparing and analyzing the ECC of different regions by simply applying foreign research results, which makes the ecological footprint model unreliable and weakens its ability as an ecosystem assessment tool. Therefore, these studies can neither accurately reflect the regional ECC nor can they provide a basis for the analysis of its spatial pattern [18,19]. Therefore, improving the process of the calculation of the conversion factor of ECC is required.

At present, the research mainly focuses on the improvement of the ECC model [20–22] and the analysis of spatiotemporal changes and driving forces [23–25]. In addition, the research on ECC has expanded from the field of ecology to other fields as well [26–28]. Most studies on ECC are based on statistical data, and spatiotemporal changes of regional ECC cannot be accurately reflected. Because of the differences in climate and land productivity among different regions, if equilibrium and yield factors of other regions are directly used within a specific region, the mechanism of variation in the local ECC cannot be accurately reflected. Therefore, the key parameters of ECC (equilibrium and yield factors) should be calculated according to the actual situation of the region to reflect the regional ECC accurately. In recent years, the development of geographic information systems (GIS) and remote sensing (RS) provide powerful tools for the study of ECC [29–31]. GIS and RS also provide abundant data [32], which makes the results more robust with accurate spatiotemporal characteristics [33].

The Yellow River is the mother river of the Chinese nation. The protection and development of the Yellow River valley has always been important for China. On September 18, 2019, the policy of "Ecological protection and high-quality development in the Yellow River Basin" was implemented in China. The Yellow River Basin in Shaanxi (YRBS) is an important ecological barrier in China. The Yellow River Basin covers the Wei River Impact Plain on the northern slope of Qin Mountains province, including sandy areas along the Great Wall, hilly and gully areas, the northern dry plateau and Guanzhong Plain, Shanbei Plateau, and the Qin Mountains. It is also the region with the most serious soil and water loss and has one of the most fragile ecological environments. The YRBS is also a key ecological management area for converting cropland to forest and it is home to many small watershed management programs. The protection and management of this region is important for the development strategy of the Yellow River Basin.

The purpose of this study is to explore the methods for evaluating the ecological assessment of the Yellow River Basin and provide suggestions for the ecological protection and high-quality development of this region. Based on RS data and GIS technology, this study aims to evaluate the evolution of the ECC of the YRBS and explore its driving

factors. The results of this study are expected to have important theoretical and practical significance for ecological restoration and sustainable development. This study is expected to serve as a reference for the study of ECC in other similar regions. This research is the first attempt to integrate remote sensing models and geographic detectors to study ECC. All of them are based on spatial data and spatial analysis, filling the method gap. This integrated method breaks through the limitations of traditional research limited by data acquisition, laying the foundation for conducting large-scale research.

## 2. Materials and Methods

### 2.1. Study Area

Shaanxi is located in the middle reaches of the Yellow River. The Yellow River Basin accounts for 65% of the province's land area and 76% of its population. The hilly areas of Shanbei are the main sources of sediments that flow into the Yellow River. A total of 40.39 million ha of land was converted from cropland to forest and grassland in the region. The control of 15.7 million ha of desertified land has reduced the annual average quantity of sediment entering the Yellow River from 830 million tons to 268 million tons. The Shaanxi section of the river basin is responsible for more than 83% and 78% of the province's industrial and domestic water consumption, respectively. Non-point source pollution and soil erosion is severe in the Wei, Yanhe, and Wuding rivers. Per capita water resources in Shaanxi total 447 m$^3$, which less than a fifth of the national average (Figure 1).

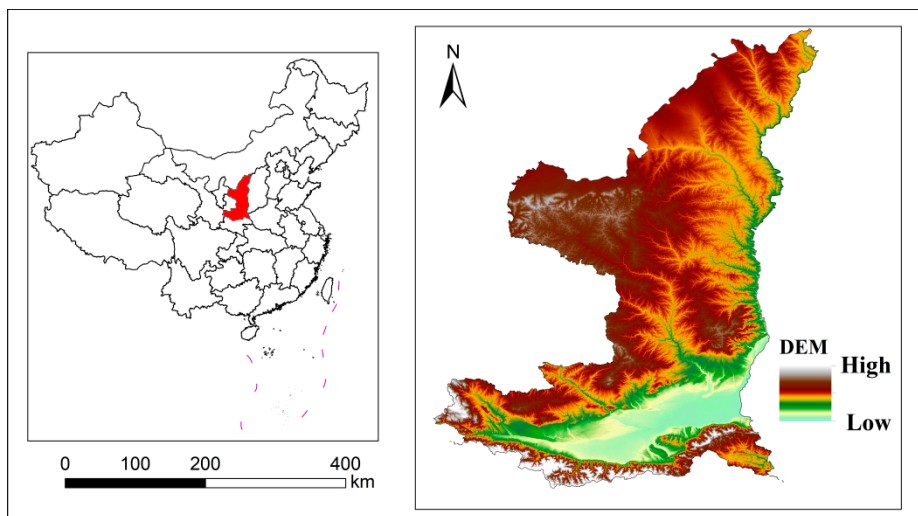

**Figure 1.** The location of the Yellow River Basin in Shaanxi.

### 2.2. Methodology

#### 2.2.1. Workflow

Based on the principles of the ecological footprint method, a calculation method that combines RS and GIS data to estimate ECC based on net primary productivity (NPP) was established. First, the model based on the Carnegie–Ames–Stanford approach (CASA) was applied to estimate the NPP. Second, the NPP of each land type was used as an indicator to reflect the productivity of the ecosystem, and the equilibrium and yield factors were determined. Then, the ECC of the watershed was calculated along with the carrying capacities of each land-use type. The spatial analysis function of GIS was used to study the spatial heterogeneity of the ECC. Finally, the geographical detector model was used to study the influencing factors of the ECC (Figure 2).

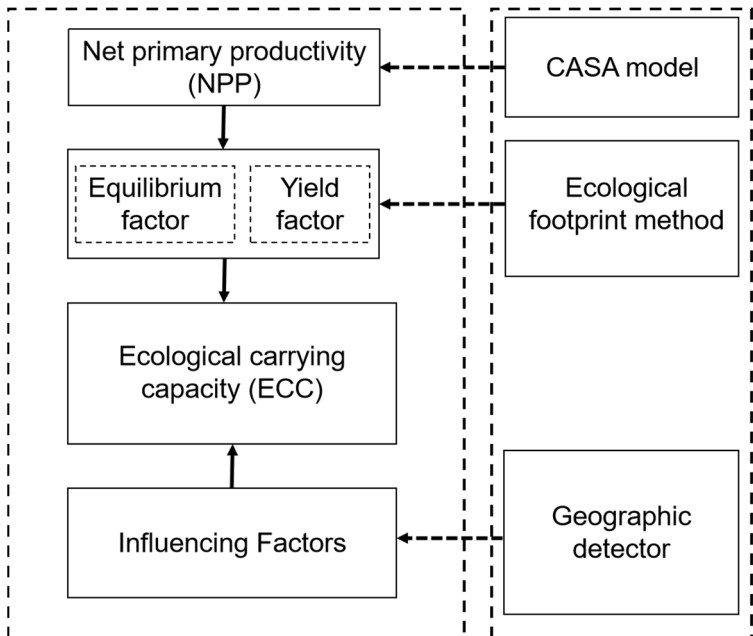

**Figure 2.** Technical route of this research.

### 2.2.2. Estimation Method for NPP

The NPP is the annual production of biomass by green plants under solar photosynthesis, which is the basis for the survival, growth, and reproduction of all consumers and decomposers on Earth. The NPP directly reflects the production capacity of plant communities under natural conditions.

This study used the modified CASA model by Wen et al. to estimate the NPP of vegetation using the following formula:

$$\text{NPP}(x,t) = \text{APAR}(x,t) \times \varepsilon(x,t) \tag{1}$$

where $\text{NPP}(x,t)$ is the accumulated total organic matter $(\text{gC}/(\text{m}^2\cdot\text{month}))$ of plants at pixel $x$ in month $t$; $\text{APAR}(x,t)$ is the effective photosynthetic radiation absorbed at pixel $x$ during month $t$ $(\text{MJ}/(\text{m}^2\cdot\text{month}))$; and $\varepsilon(x,t)$ represents the actual light utilization of plants at pixel $x$ during the month $t$.

The effective photosynthetic radiation absorbed at pixel $x$ during month $t$ is calculated as follows:

$$\text{APAR}(x,t) = \text{SOL}(x,t) \times \text{FPAR}(x,t) \times 0.5 \tag{2}$$

where $\text{SOL}(x,t)$ represents the total solar radiation at pixel $x$ for month $t$ $(\text{MJ}/(\text{m}^2\cdot\text{month}))$; $\text{FPAR}(x,t)$ is the ratio of effective photosynthetic radiation absorbed by vegetation at pixel $x$ in $t$ month $t$; and 0.5 represents the ratio of photosynthetic effective radiation to total solar radiation.

$$\varepsilon(x,t) = T_{\varepsilon 1}(x,t) \times T_{\varepsilon 2}(x,t) \times W_\varepsilon(x,t) \times \varepsilon_{\max} \tag{3}$$

where $T_{\varepsilon 1}(x,t)$ and $T_{\varepsilon 2}(x,t)$ are the influence coefficients of low and high temperature stress, respectively; $W_\varepsilon(x,t)$ is the coefficient of effect of water stress; and $\varepsilon_{\max}$ is the maximum light energy utilization rate (%) under the ideal condition.

### 2.2.3. Evaluation of ECC

In this paper, the NPP of the ecosystem is used as an important indicator to reflect the biological output of the land, which can accurately and intuitively reflect the productivity differences of various land types, and at the same time, the calculated ecological footprint can truly reflect the human impact on the ecological environment. Direct appropriation of system production and supply capacities were performed to better assess the ecological

sustainability of an area. The specific performance was evaluated using NPP to calculate the biological yield and equilibrium factors of different land-use types.

$$ECC = \sum_{i=1}^{5} A_i \times YF_i \times EQF_i \tag{4}$$

The following procedure was used to calculate various land yield factors in the Yellow River Basin of Shaanxi Province. Because of the large differences in the outputs of various biological production areas in countries or regions, the global or national yield factors were used according to the formula provided by Wackernagel et al. [34]. in terms of supply. The error in the calculation of the biological output of the regional natural system is large, and hence, the yield factor used in this paper is equal to the ratio of the NPP of a certain land-use type in the entire basin to that of the NPP in the entire country, which can better reflect the regional nature of the biological output of the natural system.

$$YF_i = \frac{NPP_i}{\overline{NPP_i}} \tag{5}$$

The equilibrium factor was calculated by dividing the NPP of a certain type of organism in the basin by the average NPP of all types of land in the basin.

$$EQF_i = \frac{NPP_i}{\overline{NPP}} \tag{6}$$

where ECC stands for ecological carrying capacity; $A_i$ represents the area of inland class $i$; $YF_i$ is the yield factor of inland type $i$; $EQF_i$ denotes the equilibrium factor of inland class $i$; $NPP_i$ denotes the average *NPP* of inland type *I*; $\overline{NPP}i$ denotes the average NPP of inland type $i$; and $\overline{NPP}$ denotes the average NPP of all land types in a basin.

### 2.2.4. Land-Use Dynamics and Land-Use Transition Matrix

The dynamic degree of land use can quantitatively express the rate of land-use change in a certain period, measure the difference of land-use change between different regions, and predict the future trend of land-use change in the region. The dynamic degree of land use includes the single dynamic degree of land use $K$ and the comprehensive dynamic degree of land use $S$. The larger the absolute value of the single dynamic degree, the faster the transformation of the land-use type. The comprehensive dynamic degree of land use represents the degree of land-use change in the study area from a macro perspective, and the larger the dynamic degree value, the more severe the degree of change. The specific calculation formulas are as follows:

$$K = \frac{U_m - U_n}{Un} \times \frac{1}{T} \times 100\% \tag{7}$$

$$S = \sum_{ij}^{n} \frac{\Delta S_{i-j}}{S_i} \times \frac{1}{T} \times 100\% \tag{8}$$

In the formula, $K$ is the dynamic degree of a certain land-use type during the research period; $U_m$ and $U_n$ are the areas of the land-use type in the study area at the beginning and end of a certain period (unit: ha), respectively; and $T$ is the research period (unit: year). $S$ represents the comprehensive dynamic degree of land in the study period; $S_{i-j}$ is the total area of the $i$-type land-use type converted to other land-use types in the $T$ period (unit: ha); $S_i$ is the initial area (unit: ha); and $T$ is the research period (unit: year).

### 2.2.5. Geographical Detector

The geographical detector model is a statistical method that is used to study the spatial heterogeneity of geographical phenomena to reveal the factors that drive heterogeneity [35,36]. The basic assumption of geographical detectors is that a study can distinguish several

subregions. When the sum of the variances of the subregions is less than the total regional variance, it indicates spatial heterogeneity. If the spatial distribution of the two variables is consistent, there is a statistical correlation between the two variables. If an independent variable has a strong influence on the dependent variable, then the spatial distribution of the independent variable and the dependent variable should be similar. This model includes four sub-detectors for factor, risk, interaction, and ecological detection. This study mainly uses factor and interaction detection.

(1)　Detection of dominant factors

Factor probing is used in research to determine the dominant factors. The factor detector calculates the q-value of each factor and quantifies the spatial variance explained by each factor. The following formula calculates factor detection:

$$q = 1 - \frac{\sum_{n=1}^{m} N_n \sigma_n^2}{N\sigma^2} \tag{9}$$

where $n = 1, 2,..., m$ is the stratification or partitioning of the independent variable $X$ and the dependent variable $Y$; $N_n$ and $N$ represent the number of units in the layer $n$ and in the entire area, respectively; and $\sigma_n^2$ and $\sigma^2$ are the dependent variables $Y$ in the layer $n$ and in the entire area, respectively. The variance of the $q$ value measures the explanatory power and ranges from 0 to 1. The larger the value of $q$, the stronger the explanatory power of the independent variable $X$ to the dependent variable $Y$ and vice versa.

(2)　Detection of interaction

The detection of interaction was used to determine whether the interaction of the independent variables $X_m$ and $X_n$ will strengthen or weaken the explanation of the dependent variable $Y$, or whether the effects of these independent variables on the dependent variable $Y$ are independent of each other. The specific method of measurement is as follows. Consider driving factors $X_1$ and $X_2$ as examples; first, calculate the explanatory power $q(X_1)$ and $q(X_2)$ of the two independent variables to the dependent variable $Y$. Second, calculate the interaction between the two independent variables and the explanatory power $q(X_1 \cap X_2)$ of the dependent variable $Y$. Third, compare the magnitudes of the three calculation results, and judge whether the influence of the interaction of the two factors on the dependent variable is enhanced or weakened relative to a single factor. The basis of judgment is shown in Table 1.

**Table 1.** Factor interaction type.

| Basis of Judgment | Interaction | Code |
|:---:|:---:|:---:|
| $q(X1 \cap X2) < min(q(X1), q(X2))$ | Nonlinear Weakened | NW |
| $min(q(X1), q(X2)) < q(X1 \cap X2) < max(q(X1), q(X2))$ | Univariate Nonlinear Weakened | UNW |
| $q(X1 \cap X2) > max(q(X1), q(X2))$ | Bivariate Enhanced | BE |
| $q(X1 \cap X2) = q(X1) + q(X2)$ | Independent | IN |
| $q(X1 \cap X2) > q(X1) + q(X2)$ | Nonlinear Enhanced | NE |

2.2.6. Selection of Driving Factors

This study refers to the results of previous studies. Considering the availability and classicality of factor data, the primary selection drivers of ECC were determined (Table 2). Combined with other factors such as weather, topography, soil, and socioeconomic variables, 11 representative indicators were selected as driving factors. These factors can better explain the forces that drive ECC at different levels. Moreover, this study also uses night light indicators as a measure to reflect the level of urban development.

**Table 2.** Representative indicators of drivers.

| Index | Code | Resolution |
|---|---|---|
| Population density | $X_1$ | 100 m |
| Average annual precipitation | $X_2$ | $0.1° \times 0.1°$ |
| Mean annual temperature | $X_3$ | $0.1° \times 0.1°$ |
| Night light data | $X_4$ | 1 km |
| Distance from highway | $X_5$ | / |
| Distance from capital city | $X_6$ | / |
| Distance from railway | $X_7$ | / |
| Elevation | $X_8$ | 30 m |
| Slope | $X_9$ | 30 m |
| Soil organic matter | $X_{10}$ | 1 km |
| Soil carbon content | $X_{11}$ | 1 km |

According to the size of the study area, a $5 \times 5$ km grid was created for reclassified data, and the pixel value of the center point of the grid was extracted. Data analysis and visualization were carried out based on the R statistical framework's geodetector package.

*2.3. Data Sources*

2.3.1. Remote Sensing Data

Land-use data were obtained from GlobeLand30 (http://www.globallandcover.com/ accessed on 16 May 2022), global land-cover data with a spatial resolution of 30 m developed by China, including for 2000, 2010, and 2020. Data on vegetation cover were obtained from the Cold and Arid Regions Scientific Data Center of the Chinese Academy of Sciences (http://westdc.westgis.ac.cn/ accessed on 18 May 2022). Data on the normalized vegetation index (NDVI) were obtained from the MODIS series products from the National Aeronautics and Space Administration (NASA) website (https://ladsweb.modaps.eosdis.nasa.gov, accessed on 20 May 2022), and they are specifically from MOD13Q1, which is a set of synthetic data for 16 days with a spatial resolution of 250 m. Night light data were obtained from the night lighting products of the National Polar-Orbiting Partnership Visible and Infrared Imager/Radiometer Suite.

2.3.2. Meteorological Data

Meteorological data were obtained from the stations of Shaanxi and its surrounding provinces. Daily meteorological station data such as maximum and minimum air temperature and average air temperature, average wind speed, sunshine hours, average relative humidity, minimum relative humidity, and precipitation were obtained from the database at http://data.cma.cn, accessed on 16 May 2022 provided by the National Meteorological Science Data Centre. The kriging interpolation method was used for spatial interpolation, which is uniformly defined as the Albers projection. Spatial raster data sets with a spatial resolution of 1 km were obtained through grid calculation, resampling, and masked extraction.

2.3.3. Other Spatial Data

Soil data were acquired from the Harmonized World Soil Database. Spatial raster data including population density and gross domestic product (GDP) were obtained from the Chinese Academy of Sciences Resource Environmental Science Data Centre (http://www.resdc.cn, accessed on 16 May 2022) with a spatial resolution of 1 km. Data on administrative boundaries and road networks were obtained from the National Geographic Information Resources Directory Service System-LRB (https://www.webmap.cn, accessed on 16 May 2022). For vector data, such as the distance to the nearest road network, common highway, highway, and railway spatial grid data with a spatial resolution of 1 km were processed in the ArcGIS 10.2 platform. Data on elevation and slope were obtained from the Shuttle Radar Topography Mission.

## 3. Results

### 3.1. Changes in the Structure of Land-Use Types in the YRBS

From 2000 to 2010, the areas of forests, shrubland, artificial ground, and bare land increased, with the largest increase in the artificial ground with an increase of 106,810 ha, an increase from 2.33% in 2000 to 3.13% in 2010 with an annual increase rate of 3.01%. Forest area increased significantly from 25.96% in 2010 to 26.22% in 2010 with an annual increase rate of 0.10%. The area of shrubland increased from 0.34% to 0.46%. The areas of cultivated land, grassland, wetlands, and waterbodies decreased. The area of cultivated land decreased the most from 39.37% in 2000 to 38.62% in 2010 with an annual decrease rate of 0.19%. The area of grassland decreased at an annual rate of 0.12%.

From 2010 to 2020, the areas of man-made surface and waterbodies increased. The largest increase was in man-made surface area from 3.13% in 2010 to 4.38% in 2020 at an annual increase rate of 3.41%. Areas of cultivated land, forests, grassland, shrubland, wetlands, and bare land decreased. Grassland area decreased the most from 29.98% in 2010 to 29.06% in 2020 at an annual decrease rate of 0.0.31%. The area of arable land decreased from 38.62% to 38.47% at an annual decrease rate of 0.04%. Wetlands decreased by 1320 ha at an annual decrease rate of 0.76% (Figure 3).

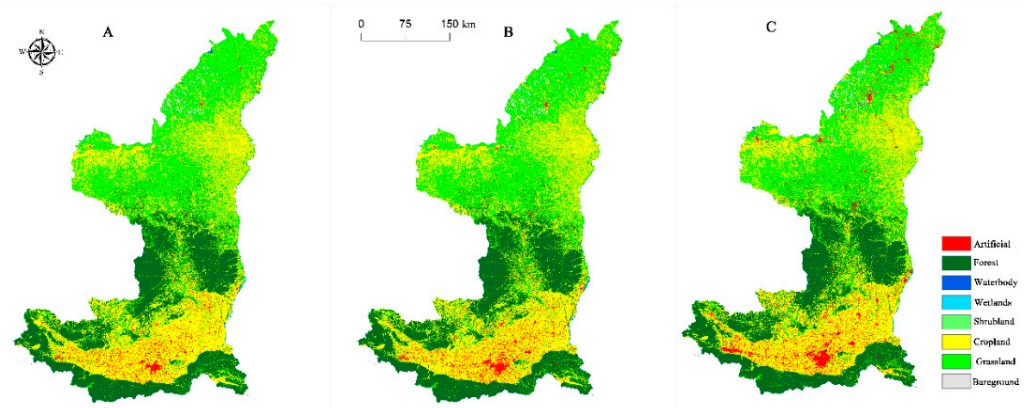

**Figure 3.** Land-cover maps of the YRBS for 2000–2020. (**A**) 2000, (**B**) 2010, and (**C**) 2020.

### 3.2. Evolution of NPP in the YRBS

Figure 4 shows the distribution of NPP in the Yellow River Basin of Shaanxi Province in 2000, 2010, and 2020. The results show that the average NPP of the entire basin in 2000, 2010, and 2020 were 203.94, 264.71, and 333.94 gC/m$^2$·year, respectively Overall, the NPP of the YRBS has increased steadily over the past 20 years. However, compared to the national average, vegetation productivity in the YRBS is low.

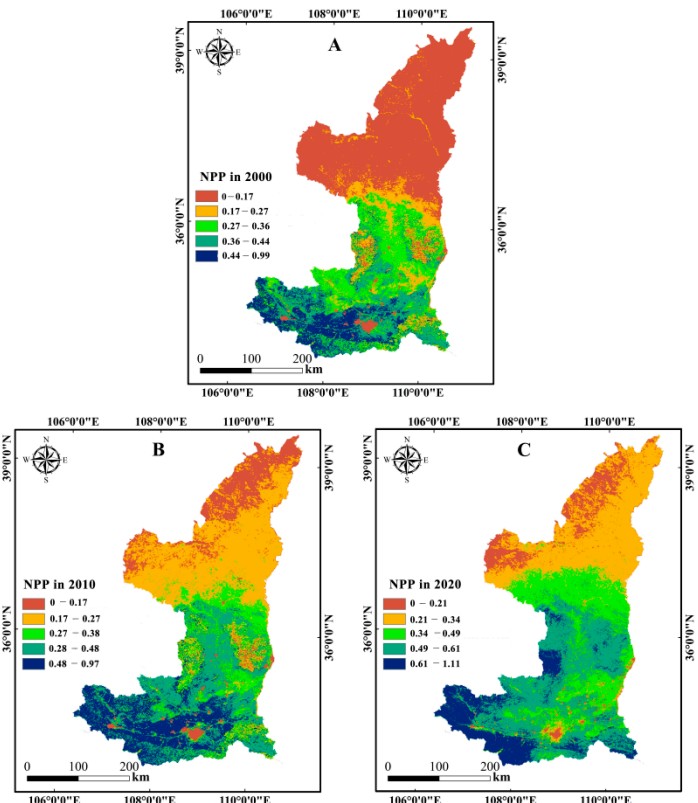

**Figure 4.** Distribution of NPP in the YRBS from 2000 to 2020 (raster mean). (**A**) 2000, (**B**) 2010, and (**C**) 2020. Unit: KgC/m$^2$·year.

### 3.3. Spatiotemporal Evolution of ECC

Figure 5 shows the spatial distribution of ECC in the YRBS in 2000, 2010, and 2020. The results show that the ECC values of the entire basin in 2000, 2010, and 2020 were $160.69 \times 10^5$, $87.64 \times 10^5$, and $147.20 \times 10^5$ wha, an overall decrease of $13.49 \times 10^5$ wha in 20 years. The spatiotemporal distribution of ECC in the YRBS shows a clear trend of first decreasing and then increasing.

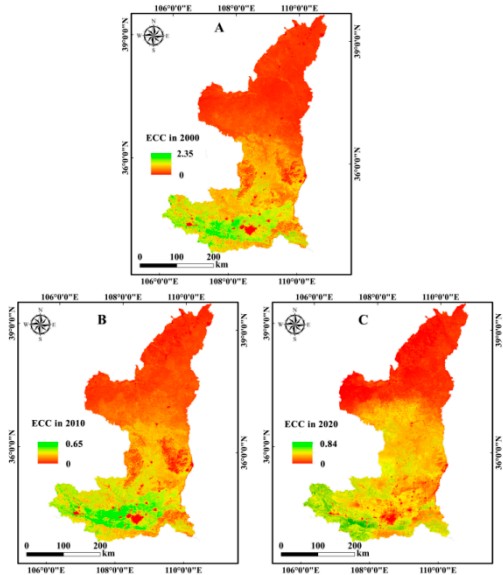

**Figure 5.** ECC in the YRBS from 2000 to 2020 (raster mean). (**A**) 2000, (**B**) 2010, and (**C**) 2020. Unit: Basin hectare (wha).

From the perspective of different land-use types (Figure 6), the ECC of cultivated land is increasing from north to south. The ECC of cultivated land showed a trend of first decreasing and then increasing over the past 20 years. The ECC of cultivated land decreased from $71.1 \times 10^5$ wha in 2000 to $39.9 \times 10^5$ wha in 2010, and then increased to $56.8 \times 10^5$ wha in 2020. The ECC of cultivated land decreased by $14.3 \times 10^5$ wha over the past 20 years. The ECC of forest land is increasing from north to south. The ECC of forest land showed a trend of first decreasing and then increasing over the past 20 years. The ECC of forest land decreased from $52.0 \times 10^5$ wha in 2000 to $24.7 \times 10^5$ wha in 2010, and then increased to $48.8 \times 10^5$ wha in 2020. The ECC of forest land decreased by $3.2 \times 10^5$ wha over the past 20 years. Grassland is mainly distributed in northern Shaanxi, and its ECC increases from north to south. The ECC of grassland showed a trend of first decreasing and then increasing over the past 20 years. The ECC of grassland decreased from $30.9 \times 10^5$ wha in 2000 to $17.6 \times 10^5$ wha in 2010, and then increased to $33.3 \times 10^5$ wha in 2020. The ECC of grassland increased by $2.47 \times 10^5$ wha over 20 years.

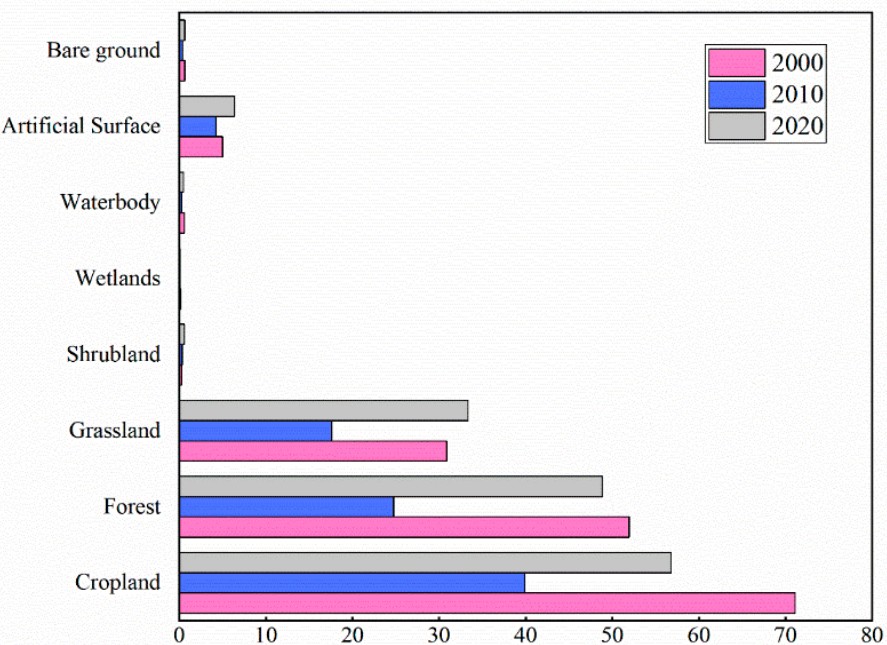

**Figure 6.** ECC for different land-use types. Unit: Basin hectare ($10^5$ wha).

### 3.4. Driving Mechanisms of ECC

#### 3.4.1. Identifying Dominant Factors

The $q$-value calculated by the geographical detector is the explanatory power of the factor. The $p$-values of all impact factors are less than 0.01, which indicates extreme significance. The larger the value of $q$, the stronger the explanatory power of the spatial distribution of the corresponding variable of the factor. The results of factor detection (Figure 7) show that the factors that have the greatest impact on the spatial differentiation of ECC are average annual precipitation, mean annual temperature, and distance from capital city, whose $q$ values are greater than 0.2.

#### 3.4.2. Interaction between Factors

The dominant factors selected through factor detection for 2000, 2010, and 2020 were used to analyze the interaction mechanisms that affect the spatial differentiation of ECC among them. The results showed that during the study period, there was a relatively close relationship between the factors (Figure 8). The $q$-values obtained from the interactions of the drivers showed varying degrees of improvement. The combined effect of these two factors will improve the explanatory power of the spatial differentiation of ECC.

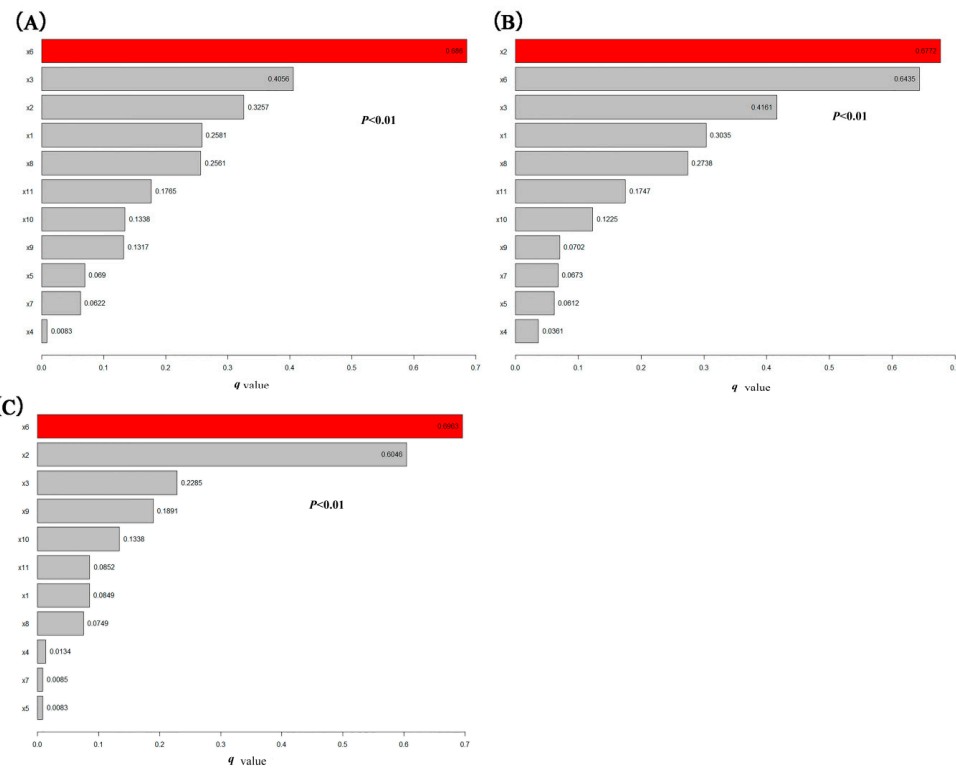

**Figure 7.** Results of factor detection (**A**) 2000, (**B**) 2010, and (**C**) 2020. (*p*-values less than 0.001 for all factors).

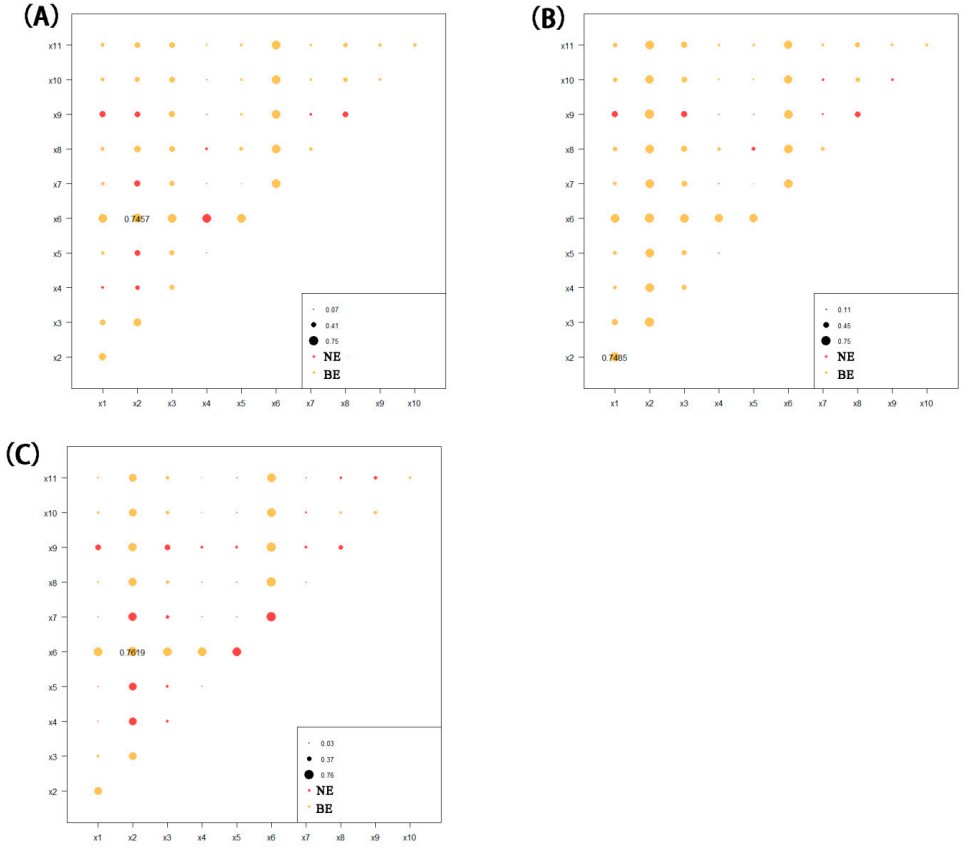

**Figure 8.** Results of the detection of interactions. (**A**) 2000, (**B**) 2010, and (**C**) 2020.

The results of the interactions show that the influence of the two-factor interaction is significantly higher than that of the single factor, and the results of the interaction of any two factors are enhanced, which indicates that the change of ECC is affected by the synergistic effect of multiple factors. The influence values of the dominant factors are shown in Figure 8. Precipitation, temperature, and distance from the provincial capital in 2000, 2010, and 2020 have the highest influence values after interaction at 0.7457, 0.7485, and 0.7619 respectively. It is evident that precipitation is the dominant factor for changes in ECC with time. The interactive explanatory power of precipitation and socioeconomic factors is increasing, indicating that the influence of human factors on changes in ECC value is gradually increasing.

## 4. Discussion

So far, there are no studies on ECC in the Shaanxi Yellow River Basin. This study integrates a new method, integrating geographic detectors and CASA models for the first time, which makes ECC analysis more convenient, and provides new research methods and understanding for ECC research.

### 4.1. ECC Evaluation Method and Data Acquisition

The variation of ECC has obvious spatiotemporal heterogeneity, and ASB data acquisition is difficult based on traditional regional survey methods. In addition, the data obtained in this way are inaccurate and lack spatiotemporal features. Satellite remote sensing data is easy to obtain, has a wide range, and is rich in information. Therefore, this research mainly uses remote sensing data combined with GIS technology to establish an index evaluation system. When evaluating the ECC, most previous studies [37–40] used statistical data and administrative divisions as units. These methods are one-sided and lack specificity. In this study, a multi-faceted and targeted evaluation method was selected. Remote sensing data are used to estimate NPP. These remote sensing data conform to the spatial scale and time frame required for research and have been used in many ecological studies [41–43]. Thus, the data are reliable. The selection of methods and data was based on a thorough investigation and understanding of the study area. The results are more in line with the actual ecological status of the ASB and are more accurate. In addition, the results have temporal and spatial characteristics, which are beneficial to analyze the reasons for changes in combination with factors such as precipitation, temperature, land use, and policies.

### 4.2. ECC Spatiotemporal Evolution and Its Determinants

Previous studies [44–46] have compared the spatial distribution of ECCs with spatiotemporal trends that mainly follow administrative divisions. This study extends from administrative zoning to natural zoning. Our approach can more accurately identify the reasons for the pattern correspondence between ECC trends and changes in drivers. The ECC of the YRBS first decreased and then increased from 2000 to 2020, and it was affected by human activities and natural factors. This is consistent with other research findings [47,48]. From the composition of ECC, it can be seen that cultivated land, forest land, and grassland have the largest variation range and are the dominant types of ECC in the YRBS. The changes in the ECC are closely related to the continuous promotion of ecological protection policies in Shaanxi Province. Shaanxi Province is one of the first provinces in China to pilot the policy of returning farmland to forests and grasslands. A series of programs for the return of farmland to forests and grasslands have enabled the restoration of forest land and grassland areas, which have relieved the stress caused by the ecological footprint to a certain extent [49,50]. To further improve the ECC of the YRBS and build a good ecological environment, it is necessary to continue the implementation of strict ecological and environmental protection policies to foster an ecologically sound civilization for the sustainable development of the basin [51–53]. Our research shows that precipitation is the dominant factor that affects the ECC of the YRBS. In addition, human activities significantly influence the ECC and is responsible for its enhancement during the past 20 years, which is

consistent with other research findings [54,55]. In the process of social development, the advancement of science and technology has led to the improvement of social productivity, which enables human beings to enhance the natural environment. With the progress of society, the improvement of population quality and the transformation of population structure will also play a role in promoting the improvement of regional ECC. For example, the improvement of education levels will enable human beings to pay more attention to the problems of the ecological environment and reduce the demand on and damage to the natural environment, thereby enhancing the regional ECC [56,57].

Few scholars have studied [58–60] the intrinsic driving mechanism of ECC spatiotemporal variation from space. Our research found that the areas with low ECC in the YRBS are distributed in northern Shaanxi, which related to the Loess Plateau region in the north and has been studied by many scholars [61–64]. However, the overall ecological risk management and control in northern Shaanxi has been in the preliminary stage over the past 20 years. Ecological restoration projects such as forest (grass) and slope farmland improvement are closely related. A series of ecological restoration projects have adjusted the land use structure to a certain extent. The area of cultivated land, forest land, and water in northern Shaanxi has increased, which has improved the ecological conditions for agricultural production. Nevertheless, problems of ecological degradation in the region persist. With the acceleration of urbanization, the demand for construction land has expanded rapidly, and human activities have exacerbated the division and occupation of cultivated land [65,66].

It is worth mentioning that we have studied the changing trend of the ECC of YRBS over the past 20 years. We found that the vegetation of the YRBS has recovered significantly and efforts toward the construction of the ecological environment has achieved remarkable results. However, the study region has a relatively fragile ecological environment, which can deteriorate when the value of ecological services applied to it is insufficient to sustain the system. In the context of global warming, extreme weather phenomena such as droughts, rainstorms, and floods have intensified, and therefore it is difficult to maintain a stable regional vegetation coverage. With economic development, the water demand for agricultural, industrial, and urban domestic needs increases, and the contradiction between regional water supply and demand will become more prominent [67–70]. Therefore, it is necessary to strengthen the assessment of the impact of climate change on regional water resources, and improve the response capabilities of key functional, restoration, and management areas to meet the needs of ecologically sound regional high-quality development.

### 4.3. Enlightenment of Sustainable Development in YRBS

The results of this study can provide valuable reference information for improving the ECC and promoting the sustainable development of terrestrial ecosystems in YRBS and other similar regions. This study found that water resources were the dominant factor affecting the ECC. Currently, global human water use exceeds sustainable levels, especially in arid and semi-arid regions [71–74]. Vegetation is ecologically important for maintaining the stability of ecosystems in these areas, but is greatly affected by water accessibility. Therefore, the measurement of water resources management and vegetation protection is crucial. Considering the differences in water stability between upstream and downstream rivers, countries should establish a better water resource cooperation mechanism to narrow the gap between upstream and downstream water resources in the river basin.

### 4.4. Limitations

In this study, the spatial expression of the ECC of the YRBS is realized on a spatial grid scale of 1 km. Although the spatial resolution is greatly improved compared to existing research, it is limited by the availability of data. In addition, the spatial resolution is still relatively low. If a higher-resolution estimation can be made, the accuracy of the results can be greatly improved and a better ECC of the watershed can be determined. Notably, the

CASA model that was used for the estimation of NPP inevitably presents errors because of the lack of field test data.

## 5. Conclusions

YRBS is a typical ecologically fragile area, which is located in the semi-arid area of China. This study integrated and developed a set of spatial methods to study the ecological carrying capacity. This method is suitable for areas similar to YRBS. We used RS and GIS technology to estimate the ECC of the YRBS based on the ecological footprint method of NPP of vegetation and used geographical detectors to study its driving factors.

The results show that the ECC of the YRBS changed significantly from 2000 to 2020, showing a pattern of low in the north and high in the south. Precipitation is the dominant factor that affects the ECC of the YRBS. The impact of human activities on ECC has increased significantly in the past 20 years. The analyses can provide insights for the ecological restoration and sustainable development in the YRBS. These findings will help formulate scientific governance and sustainable development policies for the YRBS and provide methodological references for the evaluation of ECC in similar regions.

Further research can be performed in the following aspects. Due to the unique advantages of evaluation using RS, it is immensely suitable for large-scale research. Therefore, national- and even global-scale ECC evaluations can be carried out. Analyses based on the coupling of ECC and social and economic factors can be conducted to determine the relationship between the ECC and social and economic development, which will improve the relationship between man and land, particularly in the process of urbanization.

**Author Contributions:** Conceptualization, Z.Z. and Y.F.; methodology, Z.Z. and S.L.; software, Z.M.; validation, Z.M. and Z.Z.; formal analysis, G.R. and S.L.; investigation, Z.M.; writing—original draft preparation, Z.Z.; writing—review and editing, Y.F. and G.R.; funding acquisition, Y.F. All authors have read and agreed to the published version of the manuscript.

**Funding:** This work was supported by the Shaanxi Provincial Forestry Science and Technology Innovation Program Special Project (SXLK2020–0102), and the China Association for Science and Technology 2020 Postgraduate Science Popularization Ability Improvement Project (kxyjs202034).

**Institutional Review Board Statement:** Not applicable.

**Informed Consent Statement:** Not applicable.

**Data Availability Statement:** The data that support the findings of this study are available from the corresponding author upon reasonable request.

**Conflicts of Interest:** The authors declare that they have no known competing financial interest or personal relationships that could have appeared to influence the work reported in this paper.

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
