# Peer review of "Evaluation of Ecological Carrying Capacity and Identification of Its Influencing Factors Based on Remote Sensing and Geographic Information System: A Case Study of the Yellow River Basin in Shaanxi"

_land, doi:10.3390/land11071080_

Round 1
Reviewer 1 Report
The manuscript presents an interesting study about carrying capacity. It used the ecological footprint concept and the geographic detector to analyze the changes in carrying capacity and the determinants of changes.
The manuscript can be improved by addressing the following issues.
1. The last two to three paragraphs of the introduction section need to be rewritten to improve the justification for the study. Also, more literature should be cited to indicate the gap that this study will fill.
2. A flow chart of the methodology can make the research design more understandable.
3. What about the p-values of the dominant factors? Are they all significant?
4. The authors should check figure 7. The legend symbol does not correspond with the main figure. The color of the legend symbol is black while there is no black color in the main figure.
5. Some data sources in Table 2 do not correspond with the sources in the data sources section. For example, the population density source in the table is the UN while it is the Chinese Academy in the data sources section.
6. The conclusion is too short. Can the authors make suggestions for future research?
7. The manuscript needs moderate copy-editing. The first letters in some sentences are not capitalized. For example, in the abstract, "regions. the result shows. EC of the". Also, lines 225 and 233.
The sentence in line 27 needs to be rewritten.
There is a repetition in line 398 "To coordinate and coordinate the relationship"
Line 412 "resolution of The rate is still" should be rewritten.
This list is not exhaustive.
Author Response
Response to Reviewer 1 Comments
The manuscript presents an interesting study about carrying capacity. It used the ecological footprint concept and the geographic detector to analyze the changes in carrying capacity and the determinants of changes.The manuscript can be improved by addressing the following issues.
Point 1: The last two to three paragraphs of the introduction section need to be rewritten to improve the justification for the study. Also, more literature should be cited to indicate the gap that this study will fill.
Response 1:Thanks for your valuable advice.We have rewritten the Introduction section. Added the purposeful meaning of the study and the large amount of related literature, and made a lot of adjustments to the readability of the introduction
Point 2: A flow chart of the methodology can make the research design more understandable.
Response 2:Thanks for your valuable advice.We have added a flowchart in the method section to make our technical route better understood
Point 3:What about the p-values of the dominant factors? Are they all significant?
Response 3:Thanks for your valuable advice.We have done extensive rewrites in the Methods section to better understand how we determine the dominant factors through the GeoDetector's factor detection model, they are all important
Point 4:The authors should check figure 7. The legend symbol does not correspond with the main figure. The color of the legend symbol is black while there is no black color in the main figure.
Response 4:Thanks for your valuable advice.There are only two colors in the legend to distinguish different interaction factor types. The black legend compares not the color, but the size, and the black dots of different sizes represent the values of different sizes.
Point 5: Some data sources in Table 2 do not correspond with the sources in the data sources section. For example, the population density source in the table is the UN while it is the Chinese Academy in the data sources section.
Response 5:Thanks for your valuable advice.We have rewritten the data section to unify all data sources into one section
Point 6: The conclusion is too short. Can the authors make suggestions for future research?
Response 6:Thanks for your valuable advice.We have expanded the Conclusions section and added an outlook for future research
Point 7: The manuscript needs moderate copy-editing. The first letters in some sentences are not capitalized. For example, in the abstract, "regions. the result shows. EC of the". Also, lines 225 and 233.The sentence in line 27 needs to be rewritten.There is a repetition in line 398 "To coordinate and coordinate the relationship".Line 412 "resolution of The rate is still" should be rewritten. This list is not exhaustive.
Response 7:Thanks for your valuable advice.We have corrected your question. At the same time, the full text of the manuscript has been polished by native speakers.

Reviewer 2 Report
This study used remote sensing and a geographic information system to assess the EC of the Yellow River Basin in Shaanxi (YRBS) China and identify the factors influencing it. The paper is poorly written and requires extensive rewriting in all sections:
Introduction: it is too short, include more references, cite more international studies.
Page 2 Line 53: check reference [301]
1) Methodology: include GIS software.
2) Results: You should report statistical results.
3) Discussion: it is too short, discuss your findings with previous studies. Limitation: discuss model validation.
4) Conclusion: too short! Include future research.
5) English: The paper should be edited by a native speaker.
Author Response
Response to Reviewer 2 Comments
This study used remote sensing and a geographic information system to assess the EC of the Yellow River Basin in Shaanxi (YRBS) China and identify the factors influencing it. The paper is poorly written and requires extensive rewriting in all sections:
Point1: Introduction: it is too short, include more references, cite more international studies.Page 2 Line 53: check reference [301]
Response 1: Thanks for your valuable advice.We have rewritten the Introduction section. Added the purposeful meaning of the study and the large amount of related literature, and made a lot of adjustments to the readability of the introduction
Point2: Methodology: include GIS software.
Response 2: Thanks for your valuable advice.We have rewritten the method part a lot and added a section on the introduction of the technical route to make our method part more readable and fluent in logic
Point3: Results: You should report statistical results.
Response 3: Thanks for your valuable advice.We have extensively rewritten the Results section to make the results more fully reflective of the research work and more closely aligned with the Methods section
Point4: Discussion: it is too short, discuss your findings with previous studies. Limitation: discuss model validation.
Response 4: Thanks for your valuable advice.We have extensively rewritten the Discussion section and added a substantial discussion section to give our work more depth and breadth
Point5: Conclusion: too short! Include future research.
Response 5: Thanks for your valuable advice.We have expanded the Conclusions section and added an outlook for future research
Point6: English: The paper should be edited by a native speaker.
Response 6: Thanks for your valuable advice. The full text of the manuscript has been polished by native speakers.

Reviewer 3 Report
Title: “Remote Sensing Evaluation of Ecological Carrying Capacity and Geographic Detection 2 of Influencing Factors: A Case Study of the Yellow River Basin in Shaanxi”
Written by Zhiyuan Zhu, Zhikun Mei, Shilin Li,Guangxin Ren, Yongzhong Feng
The thematic focus of the peer-reviewed manuscript, in my opinion, is quite relevant and of practical interest. However, in the current version, I cannot recommend the manuscript for publication. Authors should do a lot of work to significantly improve the quality of the submitted material, both in form and content. The abstract to the article is written very badly and it needs to be completely updated. The method used by the authors is described very poorly. It is necessary to describe it in more detail while observing a competent presentation of the material. For example, some text should be placed before the formula (2). In more detail it is necessary to consider the approach to identification of dominant factors (criteria, indicators, methodology).
Authors need to pay close attention to the text design of the article. For example, throughout the article there is no space before references, in some places sentences begin with a small letter, etc.
Author Response
Response to Reviewer 3 Comments
Point 1:The thematic focus of the peer-reviewed manuscript, in my opinion, is quite relevant and of practical interest. However, in the current version, I cannot recommend the manuscript for publication. Authors should do a lot of work to significantly improve the quality of the submitted material, both in form and content.
The abstract to the article is written very badly and it needs to be completely updated.
Response 1:Thanks for your valuable advice.We have rewritten the Abstract section to better reflect our research work
Point 2:The method used by the authors is described very poorly. It is necessary to describe it in more detail while observing a competent presentation of the material. For example, some text should be placed before the formula (2).
Response 2:Thanks for your valuable advice.Thanks for your valuable advice.We have rewritten the method part a lot and added a section on the introduction of the technical route to make our method part more readable and fluent in logic
Point 3:In more detail it is necessary to consider the approach to identification of dominant factors (criteria, indicators, methodology).
Response 3:Thanks for your valuable advice.We have done extensive rewrites in the Methods section to better understand how we determine the dominant factors through the GeoDetector's factor detection model.
Point 4:Authors need to pay close attention to the text design of the article. For example, throughout the article there is no space before references, in some places sentences begin with a small letter, etc.
Response 4:Thanks for your valuable advice.We have corrected your question. At the same time, the full text of the manuscript has been polished by native speakers.

Round 2
Reviewer 1 Report
The authors have tried to revise the manuscript based on the comments but some of the issues have not been adequately addressed.
1. The gap the research will fill is still not stated in the introduction. As hints, is this the first research to use a Geographic detector for analyzing ECC, or is it the first one to integrate a Geographic detector with other methods?
2. The authors should include numbers in the abstract statement "...first decreasing and then increasing...". What are the percentages of the decrease and the increase?
3. The p-values of the dominant factors are still not included. The results of the Geographic detector analysis are expected to include the q value (dominance of the factors) and p-value (statistical significance of the dominance). P-values can be included in Figure 7.
Reviewer 2 Report
Unfortunately, my comments have not been addressed. The discussion and conclusions must be improved. You should also include statistical results. "Discussion: Authors should discuss the results and how they can be interpreted in perspective of previous studies and of the working hypotheses. The findings and their implications should be discussed in the broadest context possible and limitations of the work highlighted".
Reviewer 3 Report
I accepted all your comments.
Author Response
Thank you very much for your constructive comments, which have greatly improved the quality of the manuscript and taught us a lot, thank you